# Liver Expression of *IGF2* and Related Proteins in *ZBED6* Gene-Edited Pig by RNA-Seq

**DOI:** 10.3390/ani10112184

**Published:** 2020-11-22

**Authors:** Haidong Zhao, Mingli Wu, Shirong Liu, Xiaoqin Tang, Xiaohua Yi, Qi Li, Shuhui Wang, Xiuzhu Sun

**Affiliations:** 1College of Animal Science and Technology, Northwest A&F University, Yangling 712100, China; zhaohaidong199212@163.com (H.Z.); wumingli1993@163.com (M.W.); 15882365640@163.com (S.L.); txq@nwafu.edu.cn (X.T.); yixiaohua0820@126.com (X.Y.); 15734011693@163.com (Q.L.); wangshuhui252@163.com (S.W.); 2College of Grassland Agriculture, Northwest A&F University, Yangling 712100, China

**Keywords:** ZBED6, IGF2, IGFBPs, liver, RNA seq

## Abstract

**Simple Summary:**

Zinc finger BED-type containing 6 (ZBED6), as a regulatory factor, has different regulatory mechanisms in animal development. The intron of *insulin-like growth factor 2* (*IGF2*) regulates the development of animal muscle and adipose by combining with the binding site of *ZBED6*. As a member of the insulin-like growth factor family, IGF2 plays an important role in embryonic growth and development, cell proliferation, muscle growth and genome imprinting. In order to further study the regulatory mechanism of *ZBED6* on *IGF2*, we detected the expression of *IGF2* and related genes in *ZBED6* single allele knockout (*ZBED6*-SKO) pig tissues and analyzed differently expressed genes of the transcriptome of *ZBED6*-SKO pig liver. The results showed that the partial knockout of *ZBED6* could affect the secretion of IGF2 in pig liver but had no significant difference at the protein level. This research provides a new idea for the interaction between IGF2 and ZBED6.

**Abstract:**

*Zinc finger BED-type containing 6* (*ZBED6*), a highly conservative transcription factor of placental mammals, has conservative interaction of insulin-like growth factor 2 (*IGF2*) based on the 16 bp binding sites of *ZBED6* on the *IGF2* sequence. IGF2 is related to embryo growth and cell proliferation. At the same time, its functions in muscle and adipose in mammals have been widely mentioned in recent studies. To further investigate the mechanism of *ZBED6* on *IGF2*, we detected the expression of *IGF2* and related genes in *ZBED6* single allele knockout (*ZBED6*-SKO) pig tissues and analyzed the transcriptome of *ZBED6*-SKO pig liver. Through RNA-seq, we captured nine up-regulated genes and eight down-regulated genes which related to lipid metabolism. The results showed that the mRNA of *IGF2* had an upward trend after the partial knockout of *ZBED6* in liver and had no significant difference in protein expression of IGF2. In summary, *ZBED6*-SKO could affect the secretion of IGF2 in pig liver and its own lipid metabolism. Our research has provided basic information for revealing the regulatory mechanism of the interaction between ZBED6 and IGF2 in mammals.

## 1. Introduction

*Zinc finger BED-type containing 6* (*ZBED6*) gene is a transcription regulatory factor newly discovered in recent years [1], which is closely related to the development of muscle and adipose and could regulate the expression of various important function genes. It is unique to placental mammals and the sequence is highly conserved [2]. Insulin-like growth factor 2 (*IGF2*) is a type of cell-like regulatory factor with insulin-like structure and widely expresses in various tissues during the embryonic stage. In adulthood, the expression of *IGF2* is significantly suppressed; it mainly synthesizes in liver and transports to various parts of body. *IGF2* has many conservative functions in mammals [3,4,5,6,7]. Its functions are mainly through the combination with insulin-like growth factor 1 receptor (IGF1R) and insulin-like growth factor 2 receptor (IGF2R), or it can compete with insulin-like growth factor binding protein (IGFBP). IGFBPs regulate the circulating levels, half-life and activity of *IGF2* in target tissues by competitively binding [1,8]. The stable expression of *IGF2* is of great significance for animal growth and health. *IGF2* disorders result in various diseases such as growth retardation, diabetes, neurodegenerative diseases, atherosclerosis, osteoporosis and cancer [1]. In animal production, the abnormal expression of *IGF2* is usually associated with animal production performance [9,10]. The transcriptional regulation of *IGF2* depends on the recognition site specific to the transcription factor, so the genetic variation of *IGF2* can regulate the growth and development of animals by increasing or decreasing its corresponding transcription factor binding site. *ZBED6* could combine the GCTCG located in the third intron segment of the *IGF2*, thereby affecting the development of muscle and adipose [7,11]. The mechanism was not only verified in mice, but the modification of the *ZBED6* binding site in the third intron of pigs’ *IGF2* by CRISPER/Cas9 also produced consistent results [12].

Currently, most studies focus on the downstream regulation of *IGF2*, while there is little research on the mechanism of the upstream regulation. In this study, we selected four *ZBED6* single allele knockout (*ZBED6*-SKO) pigs and three wild type (WT) pigs and constructed the expression patterns of *IGF2* and related genes in the liver, spleen, heart, semitendinosus and semimembranosus. The RNA-seq was used to analyze the expression of *IGF2* and related genes in the liver. Quantitative real-time PCR (qRT-PCR) and western blotting were used to quantified *IGF2* and related binding proteins in the tissues of *ZBED6*-SKO pigs and WT pigs, and hematological and blood biochemistry indices were detected. We investigated the effect of *ZBED6* knockout on liver *IGF2* metabolism, expected to provide novel insight for understanding the function of *ZBED6* and *IGF2* interaction and provide the theoretical basis for lean pig molecular breeding.

## 2. Materials and Methods

All experimental procedures were performed in accordance with the Regulations for the Administration of Affairs Concerning Experimental Animals approved by the State Council of the People’s Republic of China. The study was approved by the Institutional Animal Care and Use Committee of Northwest A&F University (permit number: NWAFAC1019).

### 2.1. Samples Collection and Hematological and Blood Biochemical Indices Analysis

*ZBED6* has only one exon. CRISPR/Cas9 was used to effectively destroy the *ZBED6* exon sequence in porcine embryonic fibroblasts (PEFs). All samples were provided by Dr. Pan (Institute of Organ Transplantation, Sichuan Academy of Medical Sciences & Sichuan Provincial People’s Hospital), including liver, spleen, heart, semitendinosus, semimembranosus of four *ZBED6*-SKO and three WT Guangxi Bama mini-pigs at 4~6 months; seven pigs were full-sib or half-sib. Samples were transported in solid carbon dioxide, frozen in liquid nitrogen and stored at −80 °C for subsequent experiments. Genotypes were reconfirmed using PCR and sequencing (Sangon Biotech Co., Ltd., Shanghai, China); the primes are listed in Table 1 and the sequencing results are shown in Appendix A.

Blood samples of three *ZBED6* knockout (*ZBED6*-KO) pigs, three *ZBED6*-SKO pigs and three WT pigs were collected by puncturing the jugular vein with a syringe, collecting the blood into a serum and EDTA vacuum tube. Hematological and blood biochemistry indices were tested by Yangling Demonstration Zone Hospital. The blood routine examination was performed, including white blood cell count (WBC), neutrophil number (NEU#), lymphocyte number (LYMPH#), monocyte number (MONO#), eosinophil number (EO#), basophil number (BASO#), neutrophil percentage (NEU%), lymphocyte percentage (LYMPH%), monocyte percentage (MONO%), eosinophil percentage (EOS%), basophil percentage (BASO%), red blood cell number (RBC), hemoglobin (HGB), hematocrit (HCT), average red blood cell volume (MCV), average red blood cell hemoglobin amount (MCH), average red blood cell hemoglobin concentrated (MCHC), red blood cell distribution width variation (RDW-CV), red blood cell distribution width standard deviation (RDW-SD), platelet number (PLT), average platelet volume (MPV), platelet distribution width (PDW), small blood (PCT), large platelet count (P-LCC), large platelet ratio (P-LCR), total protein (TP), albumin (Alb), alanine aminotransferase (ALT), aspartate aminotransferase (AST), alkaline phosphatase (ALP), triglyceride (TG), high density lipoprotein cholesterol (HDL-C), low density lipoprotein cholesterol (LDL-C) and lactate dehydrogenase (LDH).

### 2.2. RNA Isolation and Library Preparation

Total RNA was isolated via the RNAiso plus protocol (Takara Biomedical Technology Co., Ltd., Beijing, China). RNA quality were evaluated using Qubit2.0 RNA (Life, USA) and a cDNA library prepared using a Hieff NGS™ MaxUp Dual-mode mRNA Library Prep Kit for Illumina^®^ (YIASEN Biotech Co., Ltd., Shanghai, China). Libraries were quantified (Qubit2.0 DNA Assay Kit, YIASEN Biotech Co., Ltd., Shanghai, China). Among them, only the liver was submitted for sequencing (Illumina Xten, San Diego, CA, USA).

### 2.3. Primers Design and qRT-PCR

qRT-PCR was performed to verify the RNA-seq expression pattern of *IGF1*, *IGF2*, *IGF1R*, *IGF2R*, *IGFBP1*, *IGFBP2*, *IGFBP3*, *IGFBP4*, *IGFBP5*, *IGFBP6* and *IGFBP7*. *ACTB* was used as the housekeeping gene (Table 1). 1000 ng total RNA was performed to reverse transcription. SYBR green was utilized in qRT-PCR (Y480 Real-Time PCR Detection System, Roche) detection (Takara, Japan). Primer concentrations were 10 nM. Amplification protocol was 95 °C for 30 s, 50 cycles at 95 °C to denature and 60 °C for 30 s to anneal. Melt curve analysis was from 55 to 95 °C, with an increment of 0.5 °C every 5 s. Samples were run in triplicate. All data were normalized to *ACTB* and calculated with the 2^−ΔΔCt^ method [13].

### 2.4. ZBED6 SKO Efficiency in RNA Level

*ZBED6* gene-edited pig were disrupted by CRISPR/Cas9. The gene editing pig was SKO type, and it could express two types of transcripts, one was normal type and another was KO transcript, which was a frame shift mutation caused by 1 bp deletion (chro9: 64521346del “T”). The WT pig only has normal type. Two pairs of primers were the same for the amplification of the gene-editor locus in the WT pig. However, for the *ZBED6*-SKO pig, the primer *ZBED6*-1F/R was used to detect normal transcripts and *ZBED6*-KO transcripts, and the primer *ZBED6*-2F/R was used to detect normal transcripts. Two pairs of primers were referenced to the wild type respectively, and the different value between the red primer and green primer for the qRT-PCR of the *ZBED6*-SKO pig could be seen as KO efficiency. To ensure the accuracy of the amplification, the green primer was set as a primer amplification refractory mutation system (ARMS), and high specific AceTaq^®^ DNA Polymerase (Vazyme Biotech Co., Ltd., Nanjing, China) was used to enhance the reliability of the ARMS (Figure 1).

### 2.5. Western Blotting

Studied tissues were lysed with RIPA lysis buffer (1 mM MgCl2, 10 mM Tris-HCl pH 7.4, 1% Triton X-100, 0.1% sodium dodecyl sulfate (SDS), and 1% Nonidet P40 cocktail). The proteins were separated using 10% SDS polyacrylamide gel electrophoresis and transferred to cellulose membranes. The membranes were incubated overnight with the following primary antibodies: anti-IGF2 (catalog number: YN1761; Immunoway, Plano, TX, USA) and anti-β actin (catalog number: 20536-1-AP; proteintech, Chicago, IL, USA). They were then immunoblotted with secondary antibody (catalog number: RS0002; Immunoway, Plano, TX, USA). Immunoreactivity was visualized with enhanced chemiluminescence and analyzed with a Quantity One system (BioRad, Hercules, CA, USA).

### 2.6. RNA-seq Analysis and Statistical

FastQC (http://www.bioinformatics.babraham.ac.uk/projects/fastqc/) was used for quality control and filtering of raw data. Trimmomatic (http://www.usadellab.org/cms/page=trimmomatic) was used to remove reads containing adapter, reads containing ploy-N and low-quality reads to get clean reads. The clean reads were mapped to the reference genome (Sscrofa11.1) using HISAT2 tools (https://ccb.jhu.edu/software/hisat2/index.shtml). Statistics of the comparison results were produced using RSeQC (http://rseqc.sourceforge.net/). To perform differential expression analysis, differentially expressed genes (DEGs) were analyzed using the R software package DESeq according to the parameters fold change >2 and false discovery rates (FDR) <0.05. Finally, the results of expression difference analysis were visualized. Gene ontology (GO) enrichment analysis of up-regulated and down-regulated genes was performed using the topGO package (http://www.geneontology.org), and Kyoto Encyclopedia of Genes and Genomes (KEGG) pathway enrichment analysis (https://www.kegg.jp/) and euKaryotic Ortholog Groups (KOG) classification enrichment analysis (https://www.ncbi.nlm.nih.gov/COG/) were performed using the clusterProfiler package to determine the biochemical metabolic pathways and signal transduction pathways in which DEGs are mainly involved. In general, when the *p* value < 0.05, it is considered that the function is significantly enriched. DEGs with significant levels of FDR were used for GO and KEGG enrichment analysis. Moreover, the pig KEGG database was used as reference data sets for gene set enrichment analysis (GSEA) to get differently expressed gene sets. GSEA was performed using the software GSEA v4.0.3 (www.broadinstitute.org/gsea), and 333 pathways of pigs were selected in the KEGG database as the reference gene sets (https://www.kegg.jp/). The three filtering criteria for differential gene sets selection were absolute value of normalized enrichment score (NES) > 1, *p* value < 0.05 and FDR < 0.25. The chi-square test and analysis of variance (ANOVA) were used with SPSS software 18.0 (IBM, Armonk, NY, USA). Statistical testing was carried on the records (*p* < 0.05 and *p* < 0.01).

The high-throughput sequencing of the transcriptome sequence data has been saved in the NCBI Sequence Reading Archive (SRA), where the accession number of the pig liver samples is PRJNA657644 (SRR12480633-SRR12480639).

## 3. Results

### 3.1. Detection of ZBED6 Gene SKO Efficiency in Pig Tissues and Blood Routine Examinations

At DNA level, the genotypes of seven pig samples were identified through sanger sequencing, No.65, No.68 and No.103 were WT, and No.58, No.64 and No.65 and No.90 were *ZBED6*-SKO; the genotypes of all samples were the same as the record. At RNA level, a two-step qRT-PCR method was used to identify the partial knockout efficiency in five tissues, including liver (57.82%), spleen (44.06%), heart (59.71%), semitendinosus (76.15%) and semimembranosus (63.38%). Compared with the WT group, the mRNA of the *ZBED6* gene was significantly down-regulated in all tissues (Figure 1). In the blood routine examination, the KO group was higher than the WT group for the level of MONO# and MONO%, (*p* < 0.05) (Appendix A). No difference was found in the other indices between the three groups (*p* > 0.05) (Appendix A).

### 3.2. Expression of IGF2 and Related Genes in ZBED6-SKO Pig Tissues

qRT-PCR was performed to detect *IGF2* and related genes in the studied tissues. There was no significant difference in *IGF1*, *IGF2*, *IGF1R*, *IGF2R*, *IGFBP1*, *IGFBP2*, *IGFBP3*, *IGFBP4*, *IGFBP5, IGFBP6* and *IGFBP7*. It is worth noting that the expression of the *IGF2* gene in the liver of *ZBED6*-SKO pigs had a certain up-regulation (*p* = 0.06) (Figure 1). The expression levels of *IGF2* and related genes were analyzed via liver transcriptome sequencing, and no significant difference was found in related genes. Moreover, *IGFBP6* is not expressed in the liver, which can be mutually confirmed with the results of qRT-PCR (Figure 2). The total expression of the *IGFBP* family was analyzed and found that the results were also not significant between the *ZBED*-SKO group and the WT group. In the analysis of the expression of insulin receptors (INSR), it was found that the expression of *INSR* increased nearly twice in the *ZBED6*-SKO group (Figure 3). At protein level, the results showed that there is no significant difference in *ZBED6*-SKO pig tissues. Moreover, there was a large inter-individual difference of IGF2 (Figure 1).

### 3.3. Quality Evaluation of Sequencing Data

For the RNA-seq, the total reads count, total bases count and average read length of the samples are shown in Table 2. Calculation of the Q scores revealed that the quality of the sequencing data met the standards for further analysis. In addition, the GC content distribution was homogeneous. Table 2 indicates that the sequencing data were well filtered for further research.

### 3.4. DEGs and Gene Enrichment in ZBED6-SKO Pig Livers

To get further insight into the effect of *ZBED6*-SKO pig livers, differential expression gene analysis and functional enrichment analysis were performed between *ZBED6*-SKO and WT pig livers. Using the *p* value (*p* < 0.05) and FDR value (FDR < 0.05) as the different standards for analysis, we obtained 139 up-regulated genes and 97 down-regulated genes as well as 9 up-regulated genes and 8 down-regulated genes respectively (Table 3 and Figure 4). A total of 26 significant differential expression pathways were obtained, including 10 up-regulated pathways: lysosome, phospholipase D signaling pathway, glycerophospholipid metabolism, phosphatidylinositol signaling system, systemic lupus erythematosus, type I diabetes mellitus, ether lipid metabolism, primary immunodeficiency, natural killer cell mediated cytotoxicity and Th17 cell differentiation. There were 16 down-regulated pathways: nicotine addiction, P53 signaling pathway, ferroptosis, endometrial cancer, oxidative phosphorylation, cysteine and methionine metabolism, alpha-linolenic acid metabolism, amino sugar and nucleotide sugar metabolism, pancreatic cancer, chronic myeloid leukemia, carbon metabolism, bacterial invasion of epithelial cells, metabolism of xenobiotics by cytochrome P450, renal cell carcinoma, small cell lung cancer and rig-I-like receptor signaling pathway (Figure 5). Among them, phospholipase D signaling pathway, glycerophospholipid metabolism, phosphatidylinositol signaling system and ether lipid metabolism were all related to lipid metabolism. It is worth mentioning that when the standard of DEGs in GO and KEGG analysis is reduced from FDR to *P* value, the DEGs are significantly enriched to the gene sets and signaling pathways including fatty acid biosynthesis, TNF signaling pathway, ATP binding, transcription export complex, T cell differentiation in thymus, response to cytokine, I-kappaB kinase/NF-kappaB signaling, skeletal muscle cell differentiation and negative regulation of transcription from RNA polymerase II promoter, some of which are related to fat development (Appendix A). This is consistent with the results of GSEA analysis.

## 4. Discussion

High sequence conservation of the *ZBED6* takes place during the development of placental mammals [1]. The *ZBED6*-*IGF2* interaction is also conservative and based on the 16 bp binding sites of ZBED6 located in the *IGF2* sequence [1,10,12]. A mountain of research has shown that the *ZBED6*-*IGF2* axis has an important role in the development of muscle and other tissues [12,14], and the level of IGF2 in the blood was the main focus of attention in their studies. Although liver is the main organ secreting IGF2, there have been few reports about the expression of IGF2 and IGFBPs in the liver. Therefore, the purpose of this study was to explore the expression pattern of *IGF2* and their binding proteins in *ZBED6* gene-edited pig, to provide a reasonable explanation of the functions of the *ZBED6*-*IGF2* axis in liver, muscle and other tissues.

At RNA level, we analyzed the partial knockout efficiency of the *ZBED6* gene and found that the partial knockout efficiency of *ZBED6* in different tissues ranged from 44.06% to 76.15%. Therefore, we speculate that this is due to the existence of some compensation mechanisms between different tissues after *ZBED6* knockout [15]. As a member of the insulin-like growth factor family, *IGF2* is widely involved in many physiological and metabolic processes in the body and plays an important role in cancer development, neuromodulation, diseases of glucose metabolism, osteoporosis, muscle development and fat deposition [16,17]. Due to the partial knockout of ZBED6, IGF2 mRNA was up-regulated in the liver, but this was not statistically significant (*p* = 0.06). It may be caused by the gradual decline in the expression of IGF2 in adulthood and the increase in degree of methylation. In the other tissues, the expression of IGF2 was not significantly different between the *ZBED6*-SKO group and the WT group, which was caused by the lower expression level of IGF2 than liver. However, the expression of *IGF2* in the studied tissues was not changed significantly at the protein level, which may be due to the decrease of the total level of *IGFBP* expression in the liver. This phenomenon could have a certain relationship with the maintenance of *IGF2* steady state. Similar to our study, Younis found that when IGF2 expression was up to 100 fold, no significant changes were found in some phenotypes of the male samples [7]. This could be related to some unknown mechanisms or regulatory factors on IGF2 regulation. In addition, even if we chose half-sibling or full-sibling individuals, there are still large differences in the expression of IGF2 at the protein level of different individuals, which may be affected by individual differences and limited by current statistical methods. *IGF2* enters the various tissues with blood circulation to perform corresponding functions. *IGF2* mainly regulates its circulating level, half-life and activity in target tissues by combining with IGF1R and IGF2R or binding to IGFBPs [18,19,20]. IGF2 has an important relationship with insulin; IGF pathway signals could be mediated through IGF receptors or insulin receptors [21]. In this study, RNA-seq analysis of IGF2 receptors (IGF2R) and INSR found that IGF2, IGF2R and INSR were all up-regulated in the liver, but the IGFBPs were not up-regulated. Therefore, we speculate that the liver can use nearby free *IGF2* directly and act through *IGF2* receptors instead of relying on IGFBPs via autocrine patterns.

There were 17 differently expressed genes in knockout pig liver based on RNA-seq data (FDR < 0.05). Co-expression of *SUMF1* and sulfatase will result in a significant synergistic increase in enzyme activity, which can be used to treat polysulfatase deficiency [22]. *NIPSNAP1* activates and regulates central sensitization through extracellular signal-regulated kinase (ERK), thereby promoting the occurrence of inflammatory pain. It shows that *NIPSNAP1* can be used as a new therapeutic target for pathological inflammatory pain [23]. *SEC61A2* mediates the translocation and insertion of membrane proteins (including potassium channels) in the endoplasmic reticulum [24]. In the central nervous system, *VCAM1* regulates myelination of oligodendrocytes, and *VCAM1* knockout mice have reduced myelin thickness [25]. *ABCA7* can transport phospholipids, and the order of transporting phospholipids is phosphatidylcholine (PC)≥lysoPC>sphingomyelin (SM) = phosphatidylethanolamine (PE) [26]. *ANKS4B* plays a key role in intestinal epithelial cell microvilli assembly during intestinal cell differentiation [27]. AQP1 is a water-selective transport protein affecting the permeability of cell membranes [28]. CatSpere is a subunit of CatSper. The Catsper channel is unique to sperm and is essential for the hyperactivity of sperm flagella [29]. *EMR4* is a novel epidermal growth factor (EGF)-TM7 molecule [30]. *ZSWIM3* is the zinc finger chelation domain of *SWIM*, which could play a role in DNA binding and protein binding interactions [31]. Among them, *VCAM1* and *ABCA7* are both involved in lipid production and transportation [25,26]. ZBED6 can regulate cell proliferation, apoptosis, cell cycle, cell aggregation and neuronal differentiation [2,11,14], which could be related to the effect of VCAM on cell proliferation and thus delaying the differentiation of oligodendrocytes in vivo. In addition, ZBED6 also affects cell-to-cell N-cadherin connection, which participates in the exchange of materials and signal transduction with the external environment [32]. This might be related to the ability of DEGs CRACR2A and USP6NL to control endocytosis and signal transduction [33,34].

Significantly, expression genes (FDR <0.05) were used to evaluate gene functions in ZBED6 SKO pig liver, and no GO, KEGG or KOG signal pathways being enriched. GSEA can reveal many common biological pathways when single gene analysis does not find that the experimental group is associated with the control group. GSEA considers all of the expressed genes in an experiment, instead of selected genes with significant differential expression, which can preserve gene-gene correlations. This method uses the consistency of gene expression to obtain biological information in the data [35]. Among all the up-regulated pathways, there are the phospholipase D signaling pathway, glycerophospholipid metabolism, phosphatidylinositol signaling system and ether lipid metabolism related to lipid development and metabolic regulation, which shows a certain relationship with the function of *IGF2*. Phospholipase D (PLD) is a member of the superfamily of phospholipases, and phospholipases are essential for intracellular and extracellular signal transduction. Phosphatidic acid is the main metabolite of PLD in cells, which is a precursor of many lipids in the de novo pathway in cells. Moreover, phosphatidic acid can also be converted into two other biologically active lipid molecules, diacylglycerol (DAG) and lysophosphatidic acid (LPA) [36]. Glycerophospholipid is a kind of phospholipid and its metabolism has been found to play a role in the treatment of non-alcoholic fatty liver by drugs [37]. The phosphatidylinositol signaling system is a complex cell regulatory system composed of enzymes, phospholipid messengers and their binding proteins. In this pathway, extracellular signal molecules bind to G-protein-coupled receptors on the cell surface, and phospholipase C hydrolyzes 4,5-bisphosphatidylinositol (PIP2) on the plasma membrane into inositol—1,4,5—triphosphosate (IP3) and diacylglycerol (DG); the extracellular signal is converted into an intracellular signal, which reflects the importance of phosphatidylinositol metabolism in cell regulation [38]. Ether lipids are a unique class of glycerophospholipids, and their tissue distribution will be different. The highest levels of blood lipids are found in the brain, heart, spleen and white blood cells, while the intracellular lipid content in the liver is very low [39]. Mice and humans lacking ether lipid usually show myelination defects in the central nervous system and peripheral nervous system [40]. The myelin sheath is a layer of fatty tissue wrapped around the axons of certain neurons. It has an insulating effect and improves the conduction velocity of nerve impulses, as well as protecting the axons [41]. Since we use FDR < 0.05 as the standard to obtain fewer DEGs, the standard for DEGs is reduced from the FDR value to the *P* value for GO and KEGG analysis. The results showed that DEGs were enriched in fatty acid biosynthesis and skeletal muscle cell differentiation. Moreover, in the detection of blood biochemical indices, the KO group was lower than the WT group for the level of triglyceride (*p* < 0.05), which can be mutually confirmed with the results of GSEA.

In this study, we found that the partial knockout efficiency of ZBED6 in different tissues of *ZBED6*-SKO pigs was from 44.06% to 76.15%. The partial knockout of *ZBED6* increased the expression of *IGF2* in *ZBED6*-SKO pigs, but it did not reach a significant level (*p* = 0.06). This could be due to the gradual decline in *IGF2* expression in adulthood and the increase in methylation. At the protein level, there was a large inter-individual variation in the expression of IGF2, and the results between the *ZBED6*-SKO group and WT group were not significant, which could be related to the maintenance of IGF2 metabolism. Based on RNA-seq data, we performed gene enrichment analysis and GSEA in the liver of *ZBED6*-SKO pigs, and the results showed that the enriched pathways and gene sets were related to lipid development and metabolic regulation, which indicates a certain relationship with the function of IGF2 (Figure 6).

## 5. Conclusions

The DEGs in *ZBED6*-SKO pig liver is related to lipid metabolism, production and transportation. The expression of *ZBED6* is down-regulated in *ZBED6*-SKO pigs in five tissues and the expression of *IGF2* is up-regulated in the liver, indicating that the partial knockout of ZBED6 has a certain relationship with the secretion of IGF2 in the liver and its own lipid metabolism. At the level of protein, the expression of IGF2 is not significantly different in *ZBED6*-SKO pigs, which could be related to IGF2 metabolism.

## Figures and Tables

**Figure 1 animals-10-02184-f001:**
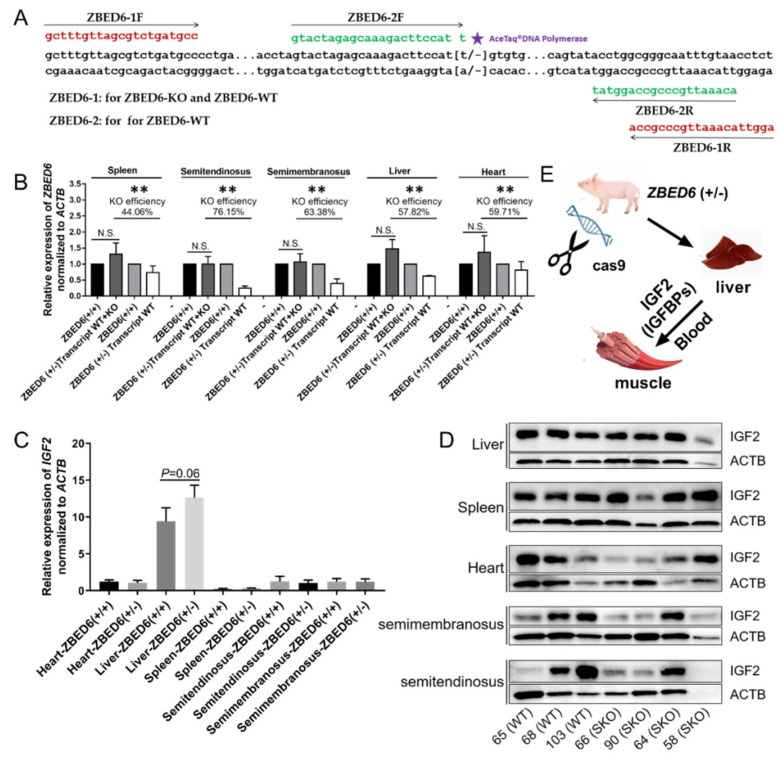
*ZBED6* and *IGF2* expression in *ZBED6*-single allele knockout (*ZBED6*-SKO) and WT pigs. (**A**) The schematic of *ZBED6* partial knockout efficiency detection; (**B**) *ZBED6* partial knockout efficiency in *ZBED6*-SKO and WT pigs in five tissues; (**C**) *IGF2* expression in *ZBED6*-SKO and WT pigs in five tissues; (**D**) Western blotting of IGF2 in *ZBED6*-SKO and WT pigs in five tissues; (**E**) the metabolism of *IGF2* in *ZBED6*-KO pigs. *ZBED6*: *zinc finger*, *BED-type containing 6*; *IGF2*: *Insulin-like growth factor 2*; KO: biallelic knockout; SKO: one allele knockout; (+/+): wild type; (+/-): single knock type. ** (*p* < 0.01).

**Figure 2 animals-10-02184-f002:**
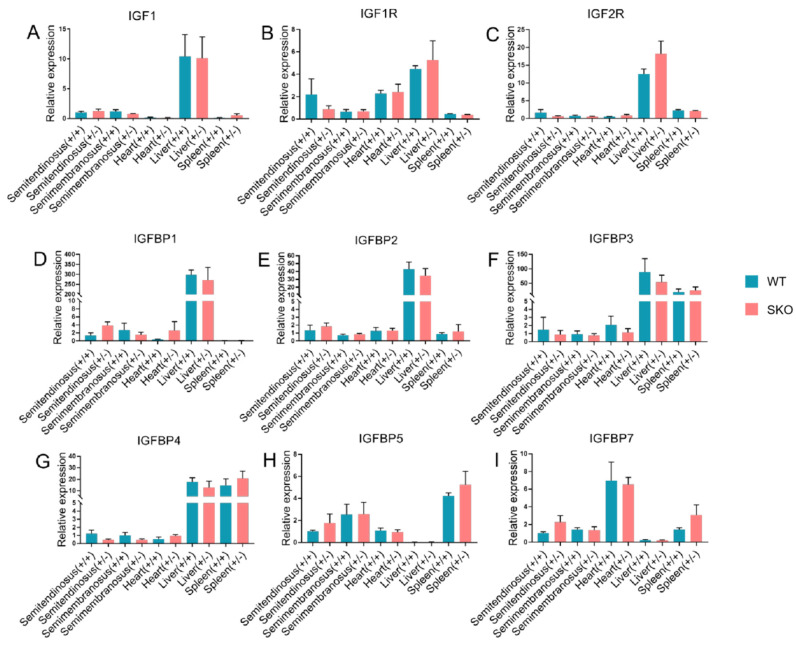
qPCR of *IGF2* and related genes expression in *ZBED6*-SKO and WT pig tissues. (**A**) The expression of *IGF1* in five tissues of *ZBED6*-SKO and WT pigs. (**B**) The expression of *IGF1R* in five tissues of *ZBED6*-SKO and WT pigs. (**C**) The expression of *IGF2R* in five tissues of *ZBED6*-SKO and WT pigs. (**D**) The expression of *IGFBP1* in five tissues of *ZBED6*-SKO and WT pigs. (**E**) The expression of *IGFBP2* in five tissues of *ZBED6*-SKO and WT pigs. (**F**) The expression of *IGFBP3* in five tissues of *ZBED6*-SKO and WT pigs. (**G**) The expression of *IGFBP4* in five tissues of *ZBED6*-SKO and WT pigs. (**H**) The expression of *IGFBP5* in five tissues of *ZBED6*-SKO and WT pigs. (**I**) The expression of *IGFBP7* in five tissues of *ZBED6*-SKO and WT pigs. Notes: IGF1: insulin-like growth factor 1; IGF1R: insulin like growth factor 1 receptor; IGF2R: insulin like growth factor 2 receptor; IGFBP 1-7: insulin-like growth factor binding protein 1-7; SKO: one allele knockout; WT: wild type; (+/+): wild type; (+/-): single knock type.

**Figure 3 animals-10-02184-f003:**
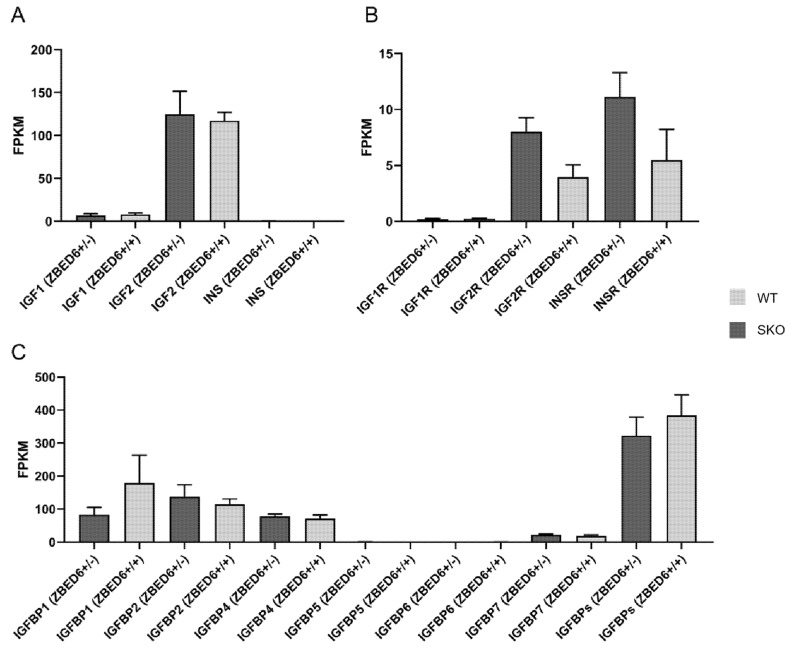
RNA-seq of IGF2 and related genes expression in *ZBED6*-SKO and WT pig livers. (**A**) The expression (FPKM) of IGF1, IGF2 and INS in *ZBED6*-SKO and WT pig livers. (**B**) The expression (FPKM) of IGF1R, IGF2R and INSR in *ZBED6*-SKO and WT pig livers. (**C**) The expression (FPKM) of IGFBP family and the total expression (FPKM) of IGFBPs in *ZBED6*-SKO and WT pig livers. Notes: SKO: one allele knockout; WT: wild type; FPKM: fragments per kilobase per million; IGF1-2: insulin-like growth factor 1-2; IGF1R: insulin like growth factor 1 receptor; IGF2R: insulin like growth factor 2 receptor; INS: insulin; INSR: insulin receptor; IGFBP1-7: insulin-like growth factor binding protein 1-7; (+/+): wild type; (+/-): single knock type.

**Figure 4 animals-10-02184-f004:**
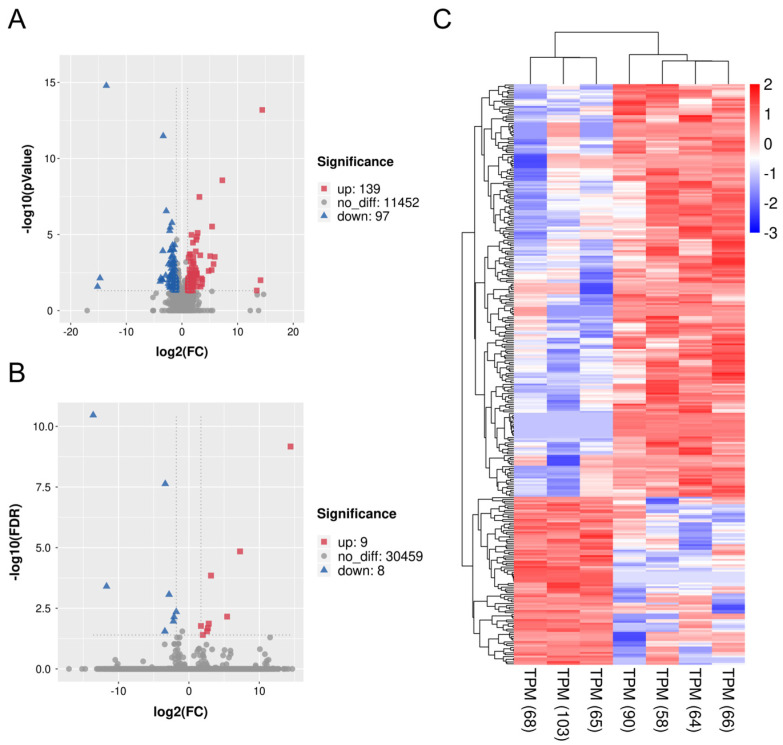
Volcano plot and heatmap of differentially expressed genes (DEGs) between *ZBED6*-SKO and WT pig livers. (**A**) Volcano plot of all significant DEGs (*P* < 0.05) between *ZBED6*-SKO and WT pig livers, including 139 up-regulated genes and 97 down-regulated genes. (**B**) Volcano plot of all significant DEGs after the FDR correction (FDR < 0.05) between *ZBED6*-SKO and WT pig livers, including 9 up-regulated genes and 8 down-regulated genes. (**C**) Heatmap of all significant DEGs (*P* < 0.05). Notes: FDR: false discovery rates; FC: fold change; TPM: transcripts per million reads.

**Figure 5 animals-10-02184-f005:**
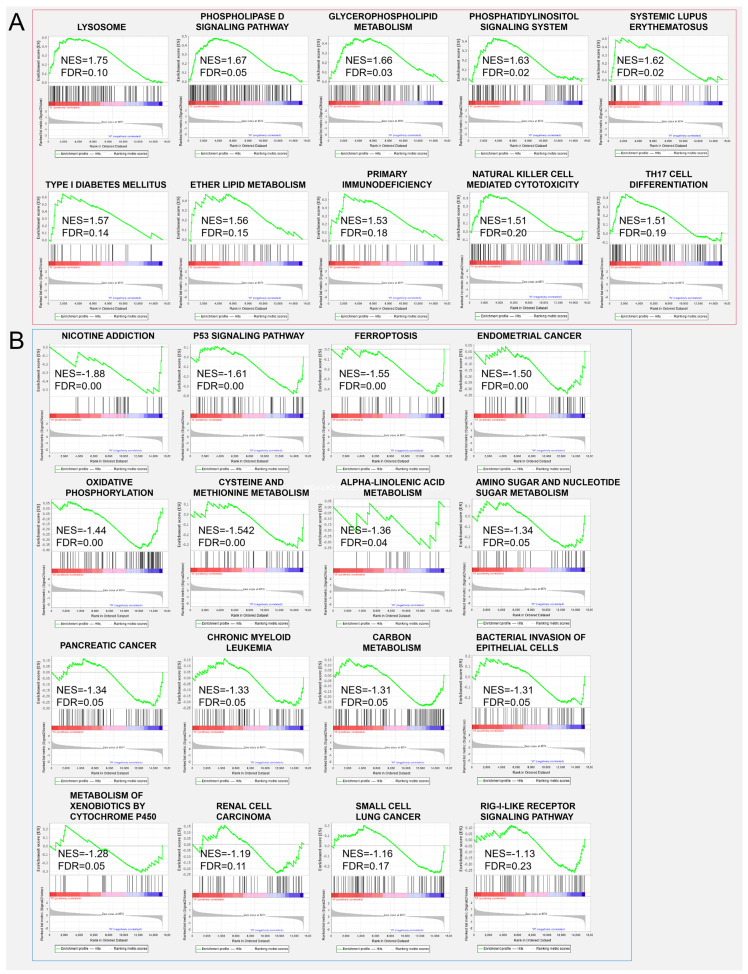
Gene set enrichment analysis (GSEA) analysis between *ZBED6*-SKO and WT pig livers (**A**) up-regulated signaling pathway enriched in *ZBED6*-SKO pig liver; (**B**) down-regulated signaling pathway enriched in *ZBED6*-SKO pig liver. Notes: NES: normalized enrichment score; FDR: false discovery rates.

**Figure 6 animals-10-02184-f006:**
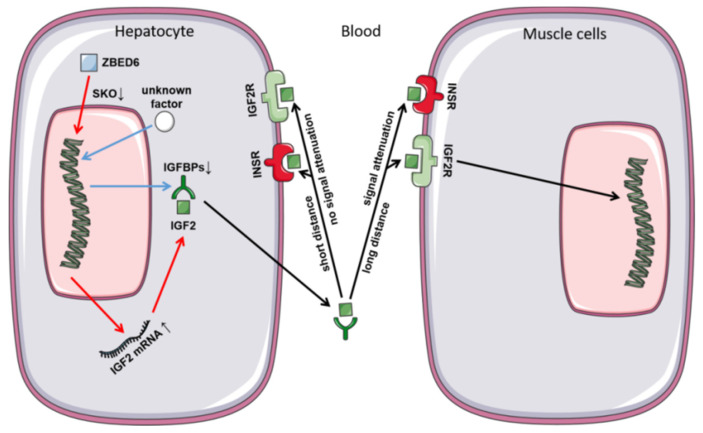
Secretion and function of IGF2 in *ZBED6*-SKO pig hepatocyte and muscle cells.

**Table 1 animals-10-02184-t001:** Primers list in this study.

Primers Name	Transcript_id	Sequence	Notes
*IGF1* ^1^	XM_021091138.1	F: GCCCAAGGCTCAGAAGGAAG	qRT-PCR
		R: GAGCAAAGGATCCTGCCAGT	
*IGF2* ^2^	XM_021080621.1	F: GTGGCATCGTGGAAGAGTGC	qRT-PCR
		R: CCAGGTGTCATAGCGGAAGAA	
*IGF1R* ^3^	XM_005659775.3	F: ACGAGTGGAGAAATCTGCGG	qRT-PCR
		R: TGAGCTTGGGAAAGCGGTAG	
*IGF2R* ^4^	NM_001244473.1	F: AGAAAGAGGTGCCGTGCTAC	qRT-PCR
		R: CCGGAGCGTGTCTATGTCTC	
*IGFBP1* ^5^	NM_001195105.1	F: ACAGCAAACAGTGCGAGACG	qRT-PCR
		R: TACTGATGGCATTTGGGGTCC	
*IGFBP2* ^5^	NM_214003.1	F: CAACCTCAAACAGTGCAAGATGT	qRT-PCR
		R: GCTGTGGTTTACTGCATCCG	
*IGFBP3* ^5^	NM_001005156.1	F: AGACGGAATACGGGCCTTGC	qRT-PCR
		R: TCTTGTCGCAGTTGGGGATG	
*IGFBP4* ^5^	NM_001123129.1	F: CAGCCCTCTGACAAGGACGA	qRT-PCR
		R: GCTCCGGTCTCGGATCTTG	
*IGFBP5* ^5^	NM_214099.1	F: CGAGCAAGCCAAGATCGAGAG	qRT-PCR
		R: AGCTTCTTTCTGCGGTCCTT	
*IGFBP6* ^5^	NM_001100190.1	F: CGCCCTCGGGGGAGAATC	qRT-PCR
		R: ATCTCAGTGTCTTGGACGCC	
*IGFBP7* ^5^	XM_003129037.5	F: TAAGCGGTGTGTGCGTGT	qRT-PCR
		R: ATGGAAGGACCTTGCTCGC	
*ZBED6-1* ^6^	NM_001166552.1	F: GCTTTGTTAGCGTCTGATGCC	qRT-PCR for *ZBED6*-KO ^8^ and *ZBED6*-WT ^9^
		R: AGGTTACAAATTGCCCGCCA	
*ZBED6-2* ^6^	NM_001166552.1	F: GTACTAGAGCAAAGACTTCCATT	qRT-PCR for *ZBED6*-WT
		R:ACAAATTGCCCGCCAGGTAT	
*ACTB* ^7^	XM_021086047.1	F: GGACTTCGAGCAGGAGATGG	qRT-PCR
		R: AGGAAGGAGGGCTGGAAGAG	

^1^*IGF1*: insulin-like growth factor 1; ^2^*IGF2*: insulin-like growth factor 2; ^3^*IGF1R*: insulin-like growth factor 1 receptor; ^4^*IGF2R*: insulin-like growth factor 2 receptor; ^5^*IGFBP1-7*: insulin-like growth factor binding protein 1-7; ^6^*ZBED6*: zinc finger BED-type containing 6; ^7^*ACTB*: actin beta; ^8^ KO: knockout; ^9^ WT: wild type.

**Table 2 animals-10-02184-t002:** Quality evaluation of sequencing data.

Type	*ZBED6* SKO Group		Wild Group	
Total Reads Count (#)	No.66	54877418	No.65	48523032
	No.90	42383082	No.68	44928968
	No.64	48377850	No.103	55037976
	No.58	54016486		
Total Bases Count (bp)	No.66	7856669548	No.65	6866998313
	No.90	6069354838	No.68	6387348025
	No.64	6912265759	No.103	7908177595
	No.58	7696200796		
Average Read Length (bp)	No.66	143.17	No.65	141.52
	No.90	143.20	No.68	142.17
	No.64	142.88	No.103	143.69
	No.58	142.48		
Q20 Bases Ratio (%)	No.66	98.83%	No.65	98.79%
	No.90	98.82%	No.68	98.70%
	No.64	98.83%	No.103	98.83%
	No.58	98.79%		
Q30 Bases Ratio (%)	No.66	95.61%	No.65	95.51%
	No.90	95.56%	No.68	95.27%
	No.64	95.64%	No.103	95.60%
	No.58	95.52%		
GC Bases Ratio (%)	No.66	50.05%	No.65	49.21%
	No.90	48.97%	No.68	50.11%
	No.64	50.33%	No.103	49.26%
	No.58	49.61%		

**Table 3 animals-10-02184-t003:** DEGs ^1^ (FDR ^2^ < 0.05, FC ^3^ > 2) of *ZBED6*
^4^-SKO ^5^ pig liver using RNA-seq.

Gene ID	Gene Name	MeanTPM ^6^ (SKO)	MeanTPM ^6^ (WT ^7^)	log2 (FC)	*p* Value	FDR Value	Notes
ENSSSCG00000049098	*-*	0.00010000	1.23588067	−13.59325183	1.65 × 10^−15^	3.45 × 10^−11^	LncRNA
ENSSSCG00000044627	*-*	2.20908400	0.00010000	14.43116066	6.46 × 10^−14^	6.76 × 10^−10^	LncRNA
ENSSSCG00000046118	*-*	0.13868175	1.42510900	−3.361222417	3.35 × 10^−12^	2.34 × 10^−8^	Pseudogene
ENSSSCG00000003986	*ZFP211L*	1.90845525	0.01244633	7.260540758	2.74 × 10^−9^	1.43 × 10^−5^	Protein coding
ENSSSCG00000011532	*SUMF1*	18.72840650	2.12422033	3.140222832	3.39 × 10^−8^	1.42 × 10^−4^	Protein coding
ENSSSCG00000051359	*NIPSNAP1*	0.00010000	0.33187667	−11.69643149	1.13 × 10^−7^	3.96 × 10^−4^	Protein coding
ENSSSCG00000011114	*SEC61A2*	1.51700350	10.70244267	−2.818643788	2.87 × 10^−7^	8.57 × 10^−4^	Protein coding
ENSSSCG00000006862	*VCAM1*	7.08961450	24.63916067	−1.797174024	1.70 × 10^−6^	4.43 × 10^−3^	Protein coding
ENSSSCG00000023121	*ABCA7*	1.06993575	0.02500733	5.419029131	3.01 × 10^−6^	6.99 × 10^−3^	Protein coding
ENSSSCG00000007850	*ANKS4B*	9.24855825	39.92152333	−2.109866385	3.43 × 10^−6^	7.18 × 10^−3^	Protein coding
ENSSSCG00000033190	*AQP1*	0.99121175	4.57049600	−2.205085543	5.73 × 10^−6^	1.09 × 10^−2^	Protein coding
ENSSSCG00000000732	*CRACR2A*	1.02969150	0.14599400	2.81823118	7.75 × 10^−6^	1.35 × 10^−2^	Protein coding
ENSSSCG00000011120	*USP6NL*	3.51777325	1.07724900	1.707310733	1.05 × 10^−5^	1.70 × 10^−2^	Protein coding
ENSSSCG00000037808	*novel gene*	417.64328400	64.70200867	2.690388832	1.41 × 10^−5^	2.11 × 10^−2^	Protein coding
ENSSSCG00000035850	*CATSPERE*	0.07047350	0.75291833	−3.417340618	2.29 × 10^−5^	2.81 × 10^−2^	Protein coding
ENSSSCG00000040016	*EMR4*	12.88731075	2.17047567	2.569868088	2.26 × 10^−5^	2.81 × 10^−2^	Protein coding
ENSSSCG00000007427	*ZSWIM3*	1.82821425	0.46398233	1.978293372	3.43 × 10^−5^	3.98 × 10^−2^	Protein coding

Notes: ^1^ DEGs: different expression genes; ^2^ FDR: false discovery rates; ^3^ FC: fold change; ^4^
*ZBED6*: zinc finger BED-type containing 6; ^5^ SKO: single allelic knockout; ^6^ TPM: transcripts per million reads; ^7^ WT: wild type.

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
