# Peer review of "Liver Expression of IGF2 and Related Proteins in ZBED6 Gene-Edited Pig by RNA-Seq"

_animals, 2020, doi:10.3390/ani10112184_

Round 1

Reviewer 1 Report

The authors improved the manuscript significantly. They made the necessary corrections according suggestions or provided the explanations.

I have no additional comments.

I believe, that the manuscript can be accepted in its present form.

Author Response

Thank you for your comments and support. There are still a few problems in our manuscript, so we have made some changes to improve the manuscript.

Reviewer 2 Report

Dear Authors, 

The manuscript was thoroughly improved comparing to the last version, information considered the manner of inactivation of one of ZBED allele was added. Moreover, information was added about how in SKO pigs ZBED6 gene is expressed.  In the material and method section still is lack of information whether investigated pigs are free of IGF2 G3072A, which also modifies the action of IGF2 inhibition by ZBED6.

discussion

I have a doubt with using words knockdown and knockout in the present study where SKO pigs were used. 

definition for both meaning 

The key difference between gene knockout and knockdown is that the gene knockout is a technique where the gene of interest is completely removed (inoperative state) to study of functions of the gene while gene knockdown is another technique where the gene of interest is silenced to investigate the role of the particular gene in a biological system.

In this cases one allele is inoperative and the second is active and produces to functional ZBED6 mRNA. Here shoudl be used the meaning "partial knockout".

line 275 - authors besides list the DEGs and their function shoudl try to find the relationship between these DEGs and ZBED6 activity. Because ZBED6 has different molecular activities and regulation of IGF2 gene is only one of them.

Author Response

Dear reviewer:

Thank you for your comments and support. According to your advice, we have made some changes to improve the manuscript. Please see the attachment. Supplements and modifications are marked in red in the manuscript.

Reviewer 3 Report

Line 27; “Using” RNA-seq.

Line 32; clarify “own steady-state”

Lines 37-57; use present tense.

Line 54; which intron?

Lines 112-114; what was the primer concentration and amount of RNA or cDNA used?

Line 121; describe where the deletion is.

Lines 122-125; what are red and green primers?

Lines 133-135; what species were the antibodies raised in? You could provide clone or catalog numbers.

Line 171; it is not clear in Figure 1b that ZBED6 expression is significantly different or what the X-axis labels mean. Figure 1 needs to be clarified to indicate number of animals per group and define (+/+) and (+/-).

Figure 4 needs a more descriptive legend.

Author Response

Dear reviewer:

Thank you for your comments and support. According to your advice, we have made some changes to improve the manuscript. Please see the attachment. Supplements and modifications are marked in red in the manuscript.

This manuscript is a resubmission of an earlier submission. The following is a list of the peer review reports and author responses from that submission.

Round 1

Reviewer 1 Report

Lines 55-59; it is not clear what the connection of IGF2 and ZBED6 is.

Lines 75-80; animals and gene edits are not sufficiently described.

Line 88; how was the blood collected?

Table 1; locations of primers and reference transcript sequences are not sufficiently identified.

Lines 120-121; mutations are not explained.

Lines 123-126; what are red and green primers? The assay for efficiency is not described very well. Perhaps if I had access to Figure 1 it would have been clearer.

Line 128; I do not see a Figure 1 or legend.

Line 129; western blotting

Line 142; what version of reference genome was used?

Results; I did not have access to any of the figures or legends. The results could be described better in the text. Overall, I think the experiment is underpowered; more animals should have been used. Very little explanation is given for results in the different tissues described.

Tables 2 and 3; it is not clear what KO and SKO refer to.

Lines 188-193; Based on p-values some of the comparisons do not seem to be different. A correction for multiple testing should be used for the number of phenotypes tested.

Lines 210-211; how many reads were collected per library?

Reviewer 2 Report

animals-923182

The manuscript seems to be interesting, but the authors used only 3-4 samples in the KO, SKO and WT groups which very weakened the statistical analysis and the significant differences between blood parameters seem to be random. Moreover, according to supplementary files, SKO pigs have INDEL only in one allele. Whether this SKO pigs showed significant differences in pig phenotype compared to WT. Why authors did not use in RNA-seq analysis KO pigs with knockout ZBED6 gene? There is no information on how to look knockout of KO pigs. In the supplementary material, there is only small information for SKO pigs in the chromatogram how looks insertion, but no information in which part of gene was introduced. Moreover no information how KO and SKO determine of ZBED6 expression, which is highly important if authors would like to show ZBED6 effect on IGF2 expression. Moreover, there is lack of information if investigated pigs are free of IGF2 G3072A, which also modifies the action of ZBED6 inhibition.

Introduction:

line 54 - {5,9} wrong citation. Did not concern information about association between animal production and IGF2 expression. please find proper citations.

line 54. here will be better if authors cite this article "A regulatory mutation in IGF2 causes a major QTL effect on muscle growth in the pig"

Material and methods

in the material and methods should be given the information of the differences between single knockout (ZBED6-SKO) and knockout pigs (ZBED6-KO) or should be given citation where it is described. The type of deactivation of ZBED6 gene is important here. However the authors in the supplementary material given only one information that ZBED6-SKO they are heterozygotes pigs with single INDEL.

There is no information about pig phenotype of ZBED6-SKO and ZBED6-KO pigs in context fat content and muscle mass.

Why authors performed the cDNA libraries for spleen, heart and skeletal muscle tissue if these libraries were not sequenced. If authors used spleen, heart and skeletal muscle RNA only for qPCR it should be mentioned.

line 109 In how many cycles and paired-end or single sequencing was performed?

line 103 Why in RNA-seq experiment only SKO and WT were included?

line 162. why authors submitted SRA file only for liver samples? if also spleen, heart and skeletal muscle samples were sequenced?

results

All Figures/pictures are missing.

Conclusions concerns ZBED6-KO pigs and the investigation was done on SKO pigs. 

Reviewer 3 Report

The current study aimed at the identification of differences in the expression of IGF2 and IGF2 related genes in five different tissues of ZBDE6 knock-out pigs as well as the search for genes, which are differently expressed in the liver tissue of knock-out and WT pigs. Seven pigs with single knock-out of ZBDE6 were selected for the study. Generally, the paper presents an interesting research, however I recommend the revision that should be resolved before its publishing.

The major comments.

I found the results of hematological and biochemical studies not relevant concerning the main subject of the manuscript. Authors did not observe significant differences in the most of studied parameters. These data ate not discussed in the manuscript. I suggest to exclude these from the main text (see comments to Results section).

I believe, that the most interesting results of the manuscript is the identification by transcriptome analysis of genes, which are differently expressed in knock-out and WT pigs. I suggest to focus on these results and consider the results of qRT-PCRs as confirmation of the results of transcriptome studies.

Several sentences or even entire paragraphs are difficult to understand. I suggest the English language correction, if the manuscript will be resubmitted.

Minor comments.

Abstract

L24 you mean “IGF2 sequence” instead on the “IGF2 sequencing”

I suggest to remove the results of biochemical studies from the abstract, because the found differences are low significant and not discussed in the manuscript.

I believe, that the most interesting and relevant results are the identifications of DEGs. I suggest to include these data in the abstract.

Introduction.

Please, include into the Introduction section the information concerning expression level of ZBDE6 in different tissues, indicating the tissues with high expression level of ZBDE6.

Materials and methods.

Please, provide the explanation, why especially these five tissues were selected for the analysis.

Did you perform and confirm the measurements of IGF2 expression in blood od knock-out and WT pigs?

Results and discussion.

L165-166 Please, exclude the non-relevant information (the IDs of pigs)

L168-170 Please, specify the meaning of percent values (the expression level in knock-out pigs comparing to the expression level in WT pigs)? Did you perform the analysis in replicates? If yes, please, give the SE values.

L172-173 You do not need to repeat the names of tissues, which were studied. Please, write “selected tissues” or “studied tissues”.

L176 Using transcriptome analysis, the expression level of mRNA was studied, but not the proteins. Please, correct the sentence.

L180-182 I did not understand the relevance of the sentence concerning low level of INS expression in liver tissues in relation to the subject of the manuscript.

L183 What does it mean “the certain increase in the expression of INSR”? Please, give the quantitative value.

L183-184 You do not need to repeat in the Result section, that the Western blotting was used to detect the expression at protein level, because this was already specified in the Materials and Methods section.

L186 Please, indicate the quantitative value for “the large individual differences”.

Section 3.3 The authors did not find the significant differences for the most of hematological and biochemical parameters between WT and knock-out pigs. The differences with the low level of significance were found for MCHC (P = 0.026) and triglycerides (P = 0.086), herewith for the last the intermediate values were observed in the WT group. I suggest to delete the section 3.3 from the result section, move the tables 2 and 3 into the supplementary materials and make a short description of the results, presented in the table 2 and 3 in the beginning of the Result section.

L214-217 Please, move these two sentences into the Materials and Methods section.

Discussion

L240 What do you mean under sequence conservation?

L242 Do you mean IGF2 sequence instead of IGF2 sequencing?

Authors suggest, that the possible explanation for the absence of significant differences in IGF2 mRNA expression in the liver tissues between knock-out and WT pigs, is that the studies were carried out on the tissues derived from the adult animals. My question is, why the authors selected for the studies adult pigs and not the growing pigs?

Conclusion

I suggest to complete the conclusion section by the information concerning identified DEGs.